# COPD Patients’ Behaviour When Involved in the Choice of Inhaler Device

**DOI:** 10.3390/healthcare11111606

**Published:** 2023-05-30

**Authors:** Sorin Bivolaru, Ancuta Constantin, Constantin Marinel Vlase, Cristian Gutu

**Affiliations:** 1Clinical Medical Department, University of Medicine and Pharmacy Carol Davila Bucharest, 030167 Bucharest, Romania; 2Department of Cardio-Thoracic Pathology, Carol Davila University of Medicine and Pharmacy Bucharest, 030167 Bucharest, Romania; 3Department of Medical Clinical, Dunarea de Jos University of Galati, 800008 Galati, Romania

**Keywords:** asthma, COPD, inhaler treatment, exacerbation, Respimat, Genuair, Breezhaler, compliance

## Abstract

Background: Inhaler therapy plays a crucial role in controlling respiratory symptoms in patients with chronic obstructive pulmonary disease (COPD). Incorrect or partially correct use of inhaler devices causes many chronic obstructive pulmonary disease (COPD) patients to continue to have respiratory symptoms due to poor drug deposition in the airways as a result of poor inhaler technique, leading to increased healthcare costs due to exacerbations and multiple emergency room presentations. Choosing the right inhaler device for each individual patient is a bigger challenge for doctors and chronic obstructive pulmonary disease (COPD) patients. The type of inhaler device and the correct inhaler technique depends on the control of symptoms in chronic obstructive pulmonary disease (COPD). Physicians treating patients with chronic obstructive pulmonary disease (COPD) play a central role in educating patients about the correct use of inhalation devices. The steps for the correct use of inhalation devices should be taught to patients by doctors in the presence of the family so that if the patient has difficulties handling the device correctly, the family can support them. Methods: Our analysis included 200 subjects divided into two groups—recommended group (RG) and chosen group (CG)—and aimed primarily to identify the behaviour of chronic obstructive pulmonary disease (COPD) patients when faced with deciding which type of inhaler device is most suitable for them. The two groups were monitored three times during the 12-month follow-up period. Monitoring required the physical presence of the patient at the investigating physician’s office. The study included patients who were smokers, ex-smokers, and/or with significant exposure to occupational pollutants, aged over 40 years diagnosed with chronic obstructive pulmonary disease (COPD), risk group B and C according to the GOLD guideline staging, and on inhaled ICS+LABA treatment, although they had an indication for LAMA+LABA dual bronchodilation treatment. Patients presented for consultation on their own initiative for residual respiratory symptoms under background treatment with ICS+LABA. The investigating pulmonologist who offered consultations to all scheduled patients, on the occasion of the consultation, also checked the inclusion and exclusion criteria. If the patient did not meet the study entry criteria, they were assessed and received the appropriate treatment, and if the study entry criteria were met, the patient signed the consent and followed the steps recommended by the investigating pulmonologist. As a result, patient entry into the study was randomised 1:1, meaning that the first patient was recommended the inhaler device by the doctor and the next patient entered into the study was left to decide for themselves which type of device was most suitable for them. In both groups, the percentage of patients who had a different choice of inhaler device from that of their doctor was statistically significant. Results: Compliance to treatment at T12 was found to be low, but compared to results previously published on compliance, in our analysis, compliance was higher and the only reasons identified as responsible for the better results were related to the selection of the target groups and the regular assessments, where, in addition to reviewing the inhaler technique, patients were encouraged to continue treatment, thus creating a strong bond between patient and doctor. Conclusions: Our analysis revealed that empowering patients by involving them in the inhaler selection process increases adherence to inhaler treatment, reduces the number of mistakes in inhaler use of the inhaler device, and implicitly the number of exacerbations.

## 1. Introduction

Today, COPD is one of the most significant and worrying public health problems in many countries. It affects more than 600 million people worldwide and causes more than 3.5 million deaths annually [1]. In Romania, the disease affects about 1.5 million people. COPD is a disease that is perfectly preventable by quitting smoking and especially by not picking up the habit of smoking. It is a progressive, debilitating, and fatal disease characterised by decreased lung function as a result of inflammation and airway obstruction and exacerbation episodes [2]. Often, COPD is complicated by exacerbations, which signify worsening of respiratory symptoms and necessitate adjustment of bronchodilator treatment schedules, administration of oral or intravenous corticosteroids and antibiotics, and hospitalizations [3]. COPD exacerbations are a defining feature of the disease and a major source of morbidity and mortality, and their prevention is a major goal of COPD management. Because of its progressive nature, COPD patients place a financial burden on healthcare systems as a result of the frequency and severity of exacerbations [4].

Inhalation therapy delivered by inhaler devices plays a pivotal role in the control of COPD symptoms, with the advantage of locally administered low doses of a substance with high therapeutic benefits and low local and systemic adverse effects [1]. The recommendation of inhaler therapy administration for disease control is summarised in a multitude of studies and publications, including the GOLD guidelines [5,6,7]. The therapeutic benefits as a result of inhaled therapy administration are accompanied by a number of disadvantages, stemming from the difficulties patients have in understanding the difference between recommended substances and the correct use of recommended inhalers [8]. 

For COPD treatment, there are a multitude of therapeutic formulae administered via inhalation, using an extremely rich palette of inhalation devices. The most commonly used inhalation devices are pMDI and DPI [9]. On the inhaler market, there are a lot of inhaler devices, but our analysis was limited to the three devices available in Romania with which the combination of a long-acting bronchodilator and long-acting muscarinic antagonist could be administered—Breezhaler, Genuair, and Respimat. Despite a very wide range of existing molecules and inhalation devices, many patients continue to have residual respiratory symptoms because of the incorrect use of these devices [1,4]. Luigino Calzetta et al. highlight the importance of regular evaluation of inhalation techniques in order to effectively control symptoms and significantly reduce the number of exacerbations [1].

Among inhaler device users, whether known to have asthma or COPD, a very large number of patients do not use their inhaler devices correctly, resulting in very poor disease control [4,5,10]. Physicians treating COPD patients play a crucial role in educating patients on the correct use of inhaler devices. Even guidelines recommend that healthcare providers demonstrate the correct use of inhaler devices when prescribing them, as well as requiring patients to demonstrate correct inhaler technique. Violaine Giurard [8], in her review published in 2011, goes even further and involves pharmacists in teaching the correct inhaler technique for inhalation, as they are the last ones to make contact with patients before dispensing COPD medications. 

Choosing the right inhaler for each individual patient is an extremely difficult task for healthcare professionals. Many patients have difficulty using their prescribed inhalers correctly, reducing the effectiveness of treatments and increasing healthcare costs because of exacerbations and multiple emergency room visits. [11]. There is a large body of research and publications that have shown that, when used correctly, there is little difference in clinical effectiveness between different types of inhalers. However, a very large percentage of asthma and COPD patients are unable to use their inhalers correctly, and the repercussions are very costly for both patients and healthcare systems [11]. There are studies that have concluded that older people and women are more likely to make mistakes in handling inhaler devices [12,13,14]. The authors of the studies went further and concluded that a higher education level correlates with a lower level of inhaler device handling errors [12,15,16]. Other studies have found that COPD patients are more likely to make mistakes compared to patients diagnosed with bronchial asthma [17].

Given the large number of asthma and COPD patients, there is very little research on inhaler device handling errors [9]. However, if we look at the issue from another perspective, that of patient involvement in the choice of inhaler device, we find that there are not many studies that have evaluated the compliance and errors of inhaler device use from this standpoint. 

According to Welche M.J. [18] and collaborators, compliance with inhaler therapy is influenced by many factors, including the patient’s understanding of the need for daily treatment. Welch et al. [18] also argue that patient involvement in the choice of an inhaler device is an important factor in effective asthma control. Improvements in compliance with inhaler therapy and reduction in the number of errors in handling inhaler devices have the potential to substantially increase treatment efficacy, reducing the frequency of exacerbations and hence healthcare costs [8,11,19]. International health experts have expressed concern that many studies tracking patient errors in the use of inhaler devices tend to exclude patients with poor inhaler technique from the outset, leading to the idea that, in day-to-day medical practice, certain elements contained in reviews and publications may not correspond to reality [11,20]. An even more worrying aspect of inhaler misuse is that, over the last 30–40 years, the range of inhalers has increased dramatically, but inhaler technique among patients has not improved significantly [11,21].

The aim of this research is to understand the behaviour of COPD patients when faced with choosing/deciding on the type of inhaler device that is best suitable for them and to identify new ways in which compliance with inhaler therapy can be increased over time. The particularity and novelty of the study is that some of the subjects were put in the position of choosing/deciding for themselves on the inhaler device and therefore the treatment they would follow during the monitoring. The main objectives of the study are:

O1: To study and understand the behaviour of COPD patients if they are involved in the choice of the inhaler device and, by default, the treatment.

O2: Empowering patients by involving them in the inhaler device(O1) selection process has a significant impact on increasing adherence to inhaler treatment, reducing the rate of inhaler device handling errors and reducing exacerbations.

The study continues with a description of the methods and tools used in Section 2, presentation of the results in Section 3, discussion in Section 4, dissemination of important findings in Section 5, recommendations in Section 6, and limitations of the study in Section 7.

## 2. Materials and Methods

The particularity of the study is that some of the subjects were put in the position of choosing/deciding for themselves on the inhaler device and, therefore, the treatment they would follow during the monitoring. Our study focused on observing the behaviour of COPD patients when involved in the choice of inhaler device. The focus was on identifying at least one error in inhaler device use, exacerbations, and compliance with treatment at the end of monitoring. 

We defined four working hypotheses:

**H1.** *The physician’s recommendation and patient’s opinion for the recommended group and physician’s opinion and patient’s choice for the chosen group on the most suitable inhalation device can be a warning signal regarding the high number of errors in inhalation device use, high frequency of exacerbations and low treatment compliance* [22,23,24].

**H2.** *Empowering patients by involving them in the inhaler device selection process, without influence from the prescriber and family, can lead to the selection of an inhaler device appropriate to the patient’s level of understanding of the steps they need to take in the correct inhaler administration of treatment* [18,19,22,24,25].

**H3.** *Incorrect use of inhaler devices is the major impediment in controlling the symptoms and improving COPD patients’ quality of life. The atypical approach by involving the patient in the choice of the inhaler device significantly reduces the number of use errors, increasing these patients’ quality of life and minimising respiratory symptomatology* [5,9,12,15,26,27,28].

**H4.** *Reducing the misuse of inhaler devices among COPD patients has a significant impact on respiratory symptoms and reducing exacerbations* [10,29,30,31,32].

Our analysis was based on the premise that empowering patients to choose their inhaler device both reduces the number of errors in inhaler device use and the frequency of exacerbations, and increases compliance with inhaler therapy.

Inclusion criteria: Patients who smoke, ex-smokers, and/or with significant exposure to occupational pollutants, over 40 years of age diagnosed with COPD risk group B and C according to GOLD guideline staging, and on inhaler treatment with ICS+LABA, although they were indicated for treatment with double bronchodilation LAMA+LABA, who reported on its own initiative to our pulmonology department for assessment, complaining of respiratory symptoms under background treatment—cough with expectoration, progressive exertional dyspnoea, and wheezing.

Exclusion criteria: no history of smoking and/or exposure to occupational hazards, radiological imaging suggestive of associated disease, HYHA III–IV stage heart failure, angina pectoris, heart rhythm disturbances, history of myocardial infarction and stroke in the last 12 months, chronic kidney disease—eGFR < 60 mL/min/1.73 mp, unbalanced DZ—HBA1c > 7.5%, II and III stage obesity—BMI > 35 kg/mp, alcohol dependence, psychiatric disease, patients with rheumatological and orthopedic pathology (difficulty in executing fine movements with upper limbs), patients with post-stroke sequelae, history of SARS-CoV-2 infection in the last 6 months and patients with major difficulties in understanding the differences between the inhaler devices presented, etc.

Patient inclusion in the study spanned 4 years (2018–2022) and the follow-up period was 12 months, with an initial follow-up at T0, an intermediate follow-up at T6, and a final follow-up at T12. Patients thus benefited from three follow-ups and three periods of training on the correct use of inhalation devices. The target group was patients already diagnosed with COPD and already on bronchodilator treatment with ICS+LABA, although according to the GOLD guideline, they would have been indicated for inhaled treatment with LAMA+LABA. Specifically, patients presented on their own initiative to the pulmonology consultation for residual or escalated respiratory symptoms under background ICS+LABA therapy. When patients presented, the pulmonologist performing the usual consultations, being also the investigating physician in the study, checked the inclusion and exclusion criteria. Thus, patients who did not meet the criteria were assessed, investigated, and then the therapeutic management was determined. Patients who met the inclusion criteria, after signing the consent form, were included in the study on a 1:1 basis, which means that the first patient entered into the study was recommended the inhaler device by the doctor and the second patient was asked to choose the inhaler device he/she considered most suitable for him/her.

The identification of eligible patients and their enrolment in the study was performed by a single investigating physician to minimise the multi-individual approach in the physician–patient relationship. All information about the study and therefore about the correct use of inhalers was provided to patients during the study by the same physician who also enrolled them in the study and monitored them during the 12 months. In this study, we defined exacerbation as a worsening of respiratory symptoms under inhaler treatment administered daily and requiring oral or intravenous corticosteroids, antibiotic therapy, and/or hospital care.

Two groups of 100 subjects were established. The first group was referred to as the recommendation group (RG)—the inhaler device was recommended to the patients by the investigating pulmonologist during the consultation. The second group was called the choice group (CG)—the patients were the ones who chose/decided on the inhaler device they considered most suitable for them and which they were going to use.

Subjects in the recommendation group were made aware that, considering their stage of disease, there are three treatment options and therefore three inhaler devices to administer the treatment—Genuair (Industrias Farmacéuticas Almirall, S.A., Barcelona, Spania), Respimat (Boehringer Ingelheim Pharma GmbH & Co. KG, Ingelheim am Rhein, Germania), and Breezhaler (Novartis Pharma GmbH, Nürnberg, Germania). After introducing the three inhalation devices and explaining the correct inhalation technique for each device, patients were asked to indicate which type of device they would have chosen if they had had the choice. The treatment and device of the patients in this group (RG) were recommended by the investigating pulmonologist. Subjects in the choice group were provided with all three inhaler devices considered in the study—Genuair, Respimat, and Breezhaler, on which occasion the steps for the correct inhaler technique were explained. Separately, the doctor noted the type of device they would choose for each patient. In the end, patients were asked to choose/decide on the inhaler device that they felt was most suitable for them. Both patients in the recommendation group and in the choice group were instructed at each follow-up visit on the correct use of the inhaler device, the need to quit smoking, and the importance of daily and continuous inhaler therapy.

IBM SPSS Statistics for Windows, Version 26.0 software was used for the statistical processing of the study data. Armonk, NY: IBM Corp. Continuous variables were analysed for normality and then expressed by mean value and standard deviation. Analysis of the association between categorical variables was performed using the cross-tabulation and χ^2^ (chi-square) test. If the results of the chi-square test were skewed enough to be disregarded Fisher’s exact test was used. To compare means according to the dichotomous variables in the study, the *t*-test for independent samples was used. One-way ANOVA followed by multiple comparisons using the Bonferroni post hoc test was used to compare means between groups. A statistical significance coefficient value of *p* < 0.05 was considered significant.

### Sample Representativity

The sample representativity rate calculated by Cochran’s sample size representation formula, based on the Z-score, indicates that in a general Romanian population of approximately 1,494,000 COPD patients, with an acceptable sampling error of 6% and an estimated assignment proportion to the sample population of 50%, with a distribution error of 10%, yields a coefficient of determination relative to the sample size of 188 persons, which means that the selected sample of 200 persons is representative for studying the behaviour of COPD patients when they are asked to choose/decide which type of inhaler is most suitable for them.

## 3. Results

Of the 200 subjects included in the analysis, 164 were men (82%) and 36 were women (18%). The mean age in the recommendation group (RG) was 63.65 years and in the choice group (CG) 65.14 years. At the first examination, most patients included in the recommendation group (32%) were users of the Diskus device, followed by those using Turbohaler (25%), Forspiro (13%), pMDI (9%), NextHaler (8%), Airmaster (7%), and Spiromax (6%). More than half of the patients (56%) had a prescription for the inhaler from their pulmonologist. A total of 18% had a prescription from their family doctor and 16% from their internist. Some 5% had a prescription from their allergologist or emergency doctor. In the choice group, most patients (35%) initially used the Diskus device, followed by Turbohaler (20%), pMDI (14%), Forspiro (12%), Airmaster (8%), Spiromax (6%), and NextHaler (5%) users. Half of the patients included in the choice group had inhaler treatment prescribed by the pulmonologist (50%). A total of 17% had a prescription from the family doctor, 13% from the internist, and 10% from the emergency doctor. Some 5% had a prescription from an allergologist and cardiologist respectively (Table 1).

Given the large number of asthma and COPD patients globally, we identified a small number of papers that looked at similarities between physician choice and patient choice regarding a particular inhaler device. In our analysis, in the recommendation group, in only 30% of cases, the patient’s choice of inhaler device was similar to the doctor’s choice, whereas in the choice group, in only 51% of cases, the patient’s choice was similar to the doctor’s choice. The difference between groups is statistically significant (χ^2^ = 9.15; df = 1.00; *p* = 0.004; Cramer’s V = 0.214) (H1). Moving on to the statistical analysis, we find that in both groups, the prescribing physician’s choice of inhaler device is significantly different from that of the patients. Common medical practice obliges the COPD patient to follow an inhaler treatment, with a device that they receive at the doctor’s recommendation. As a lot of literature reveals, the frequency of inhaler device misuse is very high. In our analysis, in the recommendation group, the distribution of patients’ choices for inhaler devices was similar, i.e., 26% for the Genuair device and 29% each for the Breezhaler and Respimat devices; within the choice group, those who opted for the Genuair device stood out at 51%, for the Breezhaler device 25%, and for the Respimat device 24% (χ^2^ = 24.88; df = 3; *p* < 0.001; Cramer’s V = 0.353) (H2). Analysis of inhaler device misuse at T12 revealed a significant difference in the percentage of misusers between the two groups. Up to 50% of those in the recommendation group made mistakes, whereas 31% of those in the choice group made mistakes (χ^2^ = 12.53; df = 2; *p* = 0.002; Cramer’s V = 0.250), the difference between the two groups being statistically significant. Given the multitude of studies published on the frequency of inhaler device use errors, our analysis highlights the importance of patient involvement in the choice of inhaler device, translated by a reduction in the frequency of inhaler errors, thus validating hypothesis H3 (Table 2).

For inhaler treatment in COPD, there is a wide range of inhaler devices that the doctor can choose from and recommend to the patient, but choosing the most suitable inhaler device for each individual patient is a difficult task. Identifying the right device for each individual patient, in our opinion, lies in the decision to actively involve the patient in the choice of inhaler device. 

Analysing the frequency of exacerbations by device type, we find that, in the recommendation group, most exacerbations were accounted for among subjects who were recommended the Breezhaler device (62.1%), then those on Genuair (57.7%), and then those on Respimat (55.2%), (χ^2^ = 10.56; df = 3; *p* = 0.014; Cramer’s V = 0.325) (Table 3).

In the choice group, most exacerbations occurred in patients who opted for the Genuair device (48.9%), then in those who opted for Breezhaler (31.1%), and then in those who opted for Respimat (20%) (χ^2^ = 1.84; df = 2; *p* = 0.399) (Table 4).

Many studies and publications have highlighted the link between errors in inhaler device use and exacerbations. Our analysis adds to the evidence already published on exacerbations. The percentage of those experiencing T12 exacerbations in the recommendation group is significantly higher (65%) compared to the percentage of those experiencing exacerbations in the choice group (45%)—(χ^2^ = 8.08; df = 1.00; *p* = 0.007; Cramer’s V = 0.201) (H4). As expected, and in agreement with data already published, in both groups, most exacerbations occurred in the elderly. In the recommendation group (RG), most exacerbations occurred in those aged 80+ years—100%, followed by the 60–69 years category—70.6%, 50–59 years—63.6%, 70–79 years—61.5%, and the fewest (25%) in those aged 40–49 years. In the choice group (CG), most exacerbations occurred in those aged 70–79 years—57.7%, followed by 50–59 years—52%, 80+—50%, 60–69 years—33.3%, and the fewest (25%) in those aged 40–49 years.

Introducing a new approach to COPD treatment by involving patients in the choice of inhaler device, combined with an increased frequency of visits to the doctor and encouraging patients to follow inhaler treatment, strengthens patient confidence in the prescribing doctor, thereby contributing to a reduction in the frequency of inhaler device misuse.

## 4. Discussion

Analysing our results and integrated among the already published evidence on the link between inhaler device use errors and exacerbations, the proportion of those with T12 exacerbations in the recommended group (RG) is significantly higher (65%) compared to the proportion of those with exacerbations in the choice group (45%) (CG), forcing us to take patient involvement in the inhaler device choice process very seriously. The results of our analysis found that in both RG and CG, the choice of the pulmonologist and the choice of patients when it comes to choosing an inhaler device are significantly different. As revealed by a lot of publications, the frequency of inhaler device misuse is very high. Mistakes can also be attributed to the lack of patient involvement in the inhaler device selection process. Our analysis identified in the choice group (CG), where the device was chosen by the patients, their preference was for the Genuair device (51%). Continuing the analysis of the results of our study with the errors in the use of inhaler devices at T12 in the card of the two groups, it is observed that in CG, 19% fewer errors were identified compared to RG. 

Inhaled administration of treatment is the most effective route due to the rapid local effects and limited systemic adverse effects [12]. Choosing the right inhaler device for each individual patient is a difficult task for practitioners, but it is an important element in the management of COPD symptoms [1,21,29]. 

Despite advances in medical engineering, the correct handling of inhalation devices remains a strategically important issue. Correct inhaler technique also depends very much on the type of inhaler chosen, as some inhalers are more prone to handling errors [12,33]. Correct inhalation technique depends on a number of factors, including chronic physical pathology (neurological, rheumatological, orthopedic diseases, etc.) and the patient’s mental pathology, as well as the patient’s dependence on smoking, alcohol, or other prohibited substances. Treatment compliance rates among COPD patients vary depending on the analysis from 22% to 78% [33,34,35,36], but it can be increased by involving patients in the choice of inhaler device and regular check-ups at intervals of 3–6 months. 

Crystyan et al. [9] in a meta-analysis stated that up to 100% of patients made an error in inhaler device handling and in 92% a critical error was identified. A similar situation was identified by Andrea S. Melani et al. [12] in their 2011 study drawing attention to the high number of critical errors in all inhaler device users. Furthermore, she identified a strong association between the number of critical errors and older age and/or low education. She also points out that poor inhaler uses leads to an increased risk of hospitalisation and admissions to hospital wards or regular visits to the doctor. However, many elements that would contribute to increasing COPD patients’ compliance with treatment remain dilemmatic [2,27,37,38,39], which the present study tries to contribute to by involving patients in the choice of inhaler device. Compliance with recommended treatment depends on a wide range of factors: age, gender, stage of disease, associated chronic diseases, society, family environment, level of education, financial condition, trust in the doctor, trust in the therapeutic formula, etc. [2,27,39,40]. In most reviews of inhaler device misuse, patients are carefully instructed on the correct use of inhaler devices, and patients who have difficulty using inhalers correctly are not included in studies. In reality, the situation is different, in that training in the correct use of inhalers very often does not take place, and if it does, it is completed on a ‘fast track’. This results in many patients not fully benefiting from the advantages of inhalation therapy because of poor training in correct inhalation techniques and errors in inhalation device use [7].

Smoking cessation should be the primary approach of clinicians when it comes to the non-pharmacological treatment of COPD, given the already existing medical evidence of significant improvement in the annual decrease in FEV1 after smoking cessation. In addition to reducing the rate of annual decline in FEV1, smoking cessation makes a significant contribution in the short and long term to improving respiratory symptoms. Another benefit of smoking cessation in addition to those already listed is an increase in the efficacy of inhalation treatment in patients who successfully quit smoking [3].

## 5. Conclusions

The aim of the present study was to identify the behaviour of COPD patients when faced with choosing/deciding on the most suitable inhaler device for them (O1). Attention was paid to how patients chose/decided on the most suitable device for them and how this evolved during the monitoring in terms of inhaler device misuse, exacerbation frequency, and compliance with inhaler treatment (O2). In order to achieve the two objectives, we formulated four working hypotheses (H1, H2, H3, H4), all of which were validated through the statistical data obtained and presented in detail in the section on results.

In our analysis, in both groups, a significant difference was identified between the patient’s choice of a particular device and the physician’s choice, and the simple decision to involve patients in the choice of inhaler device significantly reduced mistakes and exacerbation frequency compared to when the inhaler device was recommended by the physician regardless of patient preference. Error frequency in handling inhaler devices was significantly lower among subjects who had the opportunity to choose/decide on the type of inhaler device they wanted to use the type of inhaler device they would use during the study, compared to subjects where the device was recommended by the doctor. It can be seen that for all three device types, the frequency of errors is much lower among subjects who decided for themselves which type of inhaler device they were going to use during the study. Compliance with inhaler treatment involves active participation of patients in the recommended treatment [7] and, at T12, it was found to be low in both groups, but, compared to previously published results on compliance, compliance was higher in our analysis. Patients over 60 years of age were found to make the most frequent mistakes. Furthermore, these age groups also experienced the highest number of exacerbations. The results of our research are similar to those from a number of randomised trials that have shown a direct link between age, inhaler misuse, exacerbations, and increased healthcare costs in these age groups [12,14,28,37,41,42,43,44].

Our study concludes, along similar lines to Andrea S Melani [12], that incorrect handling of inhaler devices is associated with poor disease control and requires significant medical and financial resources across multiple exacerbations and admissions for examination or to emergency rooms. The results correlate with those already published, highlighting the importance of the physician in educating patients on the correct use of inhaler devices when prescribing inhaler treatment, compared to other educational methods—printed materials, presentation videos, leaflets, etc. [20,45,46,47,48].

## 6. Recommendations

Communication between prescriber and patient is an essential element in increasing compliance to inhaler treatment, as patients with poor compliance have been shown to exhibit low trust in the prescriber. Patients should be educated about the symptoms they need to look out for and about the differences between prescribed medicines. Treatment should be tailored and inhaler devices should be prescribed according to patients’ abilities and preferences. Doctors should consider involving the patient in choosing the type of inhaler device. The recommendation builds on the results already published, plus the results of our study which showed a statistically significant difference between patient choice and physician choice for a particular inhaler device (H1). It is well known that the thicker the therapeutic formulation, the more patients tend not to take all their treatment, which is why the use of inhalers with one, two, or more active substances, depending on the stage of the disease, contributes significantly to increasing compliance to inhaler treatment over time. Categories of patients suspected of having difficulty using pMDI devices should be recommended from the outset to receive inhaler treatment using inhalation chambers as recommended by many reviews and studies already published.

Patients with COPD should be assessed every 3–6 months or sooner on the correct handling of the inhaler device, daily administration of treatment, and smoking cessation. The number of errors in the use of inhaler devices could be reduced by involving patients in the choice of inhaler device, simply equipping inhaler devices with a dose counter, audio and visual inhaler control system, the device to be activated by breathing during inhalation, etc. Training family members in the correct handling of inhaler devices can help reduce inhaler device misuse and exacerbations. Pulmonologists prescribing inhaler devices should pay more attention to the remaining symptoms as a warning sign of poor use of inhaler devices or failure to administer daily treatment.

## 7. Study limitations

Our study was conducted in a single medical institution and within a single health system. The Romanian one. Patients were selected through the pneumology department of the “Dr. Aristide Serfioti” Military Emergency Hospital of Galati. The geographical area of origin of the patients included in the study was limited to four counties in Southeastern Romania—Galați, Brăila, Tulcea, and Vrancea, and this can be interpreted as not representative of the behaviour of COPD patients in Romania and worldwide. Although there are a lot of inhalation devices in the world, our analysis was limited to the three devices available in Romania that could be compensated by the public health insurance system and with which the combination of a long-acting bronchodilator and long-acting muscarinic antagonist could be administered—Breezhaler, Genuair, and Respimat. Our findings on the evaluation of inhaler technique may contain subjective elements as to what is meant by the errors of use of inhaler devices. Concerning exacerbations, we only validated situations in which patients enrolled in the study who sought our help for escalated or new-onset respiratory symptoms and required corticosteroid and antibiotic therapy, but we cannot completely exclude the use of antibiotics at home on their own initiative or on the recommendation of other general practitioners or pulmonologists in the patients’ home area, even if anamnestically they refuted this.

The study represents our experience collected from patients under our care and was based on the patient’s honesty and sincerity regarding the daily administration of treatment. Moreover, in our study, as in many other studies, certain categories of patients were excluded from the analysis who presented major difficulties in the use of inhalation devices from the beginning—alcohol dependence, psychiatric diseases, rheumatological and orthopaedic conditions, patients with stroke sequelae, limited financial possibilities, etc. The exclusion category also includes patients who presented major difficulties in understanding the differences between the active substances contained in inhalation devices (controller and rescue or as-needed drugs). Given that much of the patient enrolment and follow-up period coincided with the COVID-19 pandemic, and measures to limit the spread of COVID-19 limited [49] access for patients accompanied by caregivers could be a minus in the absence of family support for the correct inhalation technique education process, but we compensated for this by providing patients with correct inhalation technique educational videos for each device via the online YouTube platform. The small size of the groups analysed can be interpreted as insignificant and may reduce the importance of our analysis. Last but not least, the results should be interpreted with caution in the context of other health systems and nationalities.

## Figures and Tables

**Table 1 healthcare-11-01606-t001:** Comparison of prescription frequency charts by specialty of the original prescribing physician.

Speciality ^1^	RG Frequency	CG Frequency	Difference
Allergologist	5	5	0%
Family doctor	18	17	1%
Internist	16	13	3%
Emergency doctor	5	10	−5%
Pulmonologist	56	50	6%
Cardiologist	0	5	−5%
Total	100	100	0%
Allergologist	5	5	0%

^1^ Source: Analysis of our subjects.

**Table 2 healthcare-11-01606-t002:** Analysis of inhaler device misuse at T12.

Group ^1^	Type	Incidence of Errors at T12			Total
		N/A	Yes	No	
Recommendation Group	Frequency	14	50	36	100
	% of Group	14.0%	50.0%	36.0%	100.0%
Choice Group	Frequency	8	31	61	100
	% of Group	8.0%	31.0%	61.0%	100.0%
Total	Frequency	22	81	97	200
	% of Group	11.0%	40.5%	48.5%	100.0%

^1^ Source: analysis of our subjects.

**Table 3 healthcare-11-01606-t003:** Recommended device * exacerbation at T12.

Device Type			Exacerbation at T12		Total
			No	Yes	
Recommended Device	Non- applicable	Frequency	0	16	16
		%	0.0%	100.0%	100.0%
	Breezhaler	Frequency	11	18	29
		%	37.9%	62.1%	100.0%
	Genuair	Frequency	11	15	26
		%	42.3%	57.7%	100.0%
	Respimat	Frequency	13	16	29
		%	44.8%	55.2%	100.0%
Total		Frequency	35	65	100
		%	35.0%	65.0%	100.0%

Source: analysis of our subjects. * Analysis of the frequency of exacerbations according to the device recommended by the investigating pulmonologist (recommended group). The 16 patients for whom this does not apply represent patients who met the conditions for entry into the study, but who did not accept the change of device and initial treatment at T0, but later reversed their decision and requested the new treatment and therefore the new inhaler device.

**Table 4 healthcare-11-01606-t004:** Chosen device * exacerbation at T12.

Device Type			Exacerbation at T12		Total
			No	Yes	
Chosen Device	Breezhaler	Frequency	11	14	25
		%	20.0%	31.1%	25.0%
	Genuair	Frequency	29	22	51
		%	52.7%	48.9%	51.0%
	Respimat	Frequency	15	9	24
		%	27.3%	20.0%	24.0%
Total		Frequency	35	55	45
		%	35.0%	100.0%	100.0%

Source: analysis of our subjects. * Analysis of the frequency of exacerbations in patients in the chosen group, where the device was chosen/decided by the patients. According to the 1:1 ratio, there were no patients in the chosen group who refused at time (T) the new device and therefore the new treatment.

## Data Availability

The databases with non-nominal statistical data obtained from patients accepted for analysis may be available on request and will be detailed in the Ph.D. thesis on this topic: Comparison Between Inhaler Devices Used in COPD Treatment, authored by Sorin Bivolaru.

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
