# Peer review of "COPD Patients’ Behaviour When Involved in the Choice of Inhaler Device"

_healthcare, 2023, doi:10.3390/healthcare11111606_

Round 1
Reviewer 1 Report
I think it is important to stress the difference between a metered dose inhaler and a dry powder inhaler.
Why were nonsmokers excluded in the initial criteria?we know that not all smoking patients are affected by COPD
Please describe statistical methods in a separate section.
I think it is a retrospective study. Who was the main prescriber of ICS-LABA, the pulmonary specialist or the general practitioner?
The study should mention the effects of extrafine and alveolar deposition of the powder inhalator . The importance of a spacer device should be stated.
Please provide an explanation on how the device can be associated with exacerbation rate
Please include a legend in table 1
I suggest to include the following references useful for discussion about COPD, smoking, bronchodilation
-J Allergy Clin Immunol. 1997 Jun;99(6 Pt 1):853-4.
-Clin Respir J. 2020 Jan;14(1):29-34.
-J Taibah Univ Med Sci. 2023 Jan 12;18(4):860-867.
-J Clin Med. 2022 Dec 28;12(1):234. doi: 10.3390/jcm12010234.
-Respir Med. 2023 Feb;207:107097.
Reviewer 2 Report
The objective of the study was to compare the impact of involving COPD patients in the choice of their inhaler device versus being recommended an inhaler device based on their initial device. The study found that patients who were involved in the choice of inhaler device had higher compliance rates, fewer errors in handling their inhaler devices, and different exacerbations rate compared to those who were recommended a device.
However, the study had several limitations that need to be considered when interpreting the results. The small sample size of enrolled COPD patients may limit the generalizability of the findings. Additionally, there were several confounding factors in the study, including variations in the severity of COPD among patients, differences in the effects of various inhaler devices (such as Diskus device, Turbohaler, Forspiro, pMDI, NextHaler, Airmaster, and Spiromax), and the involvement of different physicians with varying levels of professional expertise in medical decision-making. Furthermore, it is unclear whether all patients received the same training program for their device, and the study lacked information about pulmonary rehabilitation.
Overall, the study provides evidence that involving patients in the choice of their inhaler device may be beneficial. However, due to the limitations of the study, it is difficult to draw definitive conclusions. Further research with larger sample sizes and better-controlled confounding factors is necessary to provide more robust evidence. Therefore, it is important to exercise caution when interpreting the results of this study.
Author Response
Regarding the confounding factors mentioned, the authors make the following points:
- In the inclusion criteria section, we completed that patients with COPD risk group B and C according to the Gold guideline, on ICS+LABA treatment, were included in the study, even though they had an indication for double bronchodilation LAMA+LABA treatment.
- Regarding the devices mentioned as a confounding factor - Diskus, Turbohaler, Forspiro, pMDI, NextHaler, Airmaster and Spiromax), we make the following clarification: in Romania, these devices are used to administer the ICS+LABA treatment combination and are the devices that patients were using at the time of the examination (Table 1). In conclusion, these devices were not the subject of our study. After verification of inclusion criteria and patient consent, patients received LAMA+LABA dual bronchodilation treatment administered with one of the three inhaler devices we analysed - Breezhaler, Genuair and Respimat. These three devices being the subject of our study.
- Concerning the "involvement of different physicians with different levels of professional expertise in the medical decision-making process", we make the following clarification: The identification of eligible patients and their inclusion in the study was done by a single investigating physician in order to minimize the multi-individual approach in the relationship between physician and patients. All patients received the same training programme on the correct use of the inhaler device. The training was done by the investigating physician who prescribed the treatment and who also did the monitoring during the study. (Material and method, row).
- Regarding pulmonary rehabilitation, the authors make the following points: Unfortunately, in Romania there is an extremely small number of respiratory rehabilitation centres. In the south-eastern area of Romania where the study was carried out there is no such facility, therefore patients were not able to benefit from a respiratory rehabilitation programme. Instead, all patients were encouraged to quit smoking, to take their inhaler treatment daily and to include 3-4 days of 30-minute walking in their weekly schedule. Sorin Bivolaru has also published an analysis of the benefits of physical activity: The Role of Physical Deconditioning in Cardiological and Pulmonological Medical Practice - https://doi.org/10.2478/inmed-2022-0196
Thank you!
Reviewer 3 Report
The manuscript “COPD patients’ behaviour when involved in the choice of inhaler device” aim primarily to identify the behaviour of COPD patients when faced with decision about which type of inhaler device is most suitable for them (abstract) and another objective that is to identify new ways in which compliance to inhaler therapy can be increased over time.
One of weakness aspects is that this study does not report the necessary inclusion of the family to support the patients in inhaler device use, helping them to minimize the frequency of errors and promote the compliance to this treatment.
In general, the manuscript is clear and relevant for the field of individual health, the self-care about chronic conditions, particularly respiratory diseases.
There are aspects in the manuscript that can be improved:
Abstract: In background, it is important to define more clearly the problem, not only because it is difficult for doctor to choose the inhaler devices. At the end of this topic must be the two main objectives of the study (choose and compliance about inhaler use). Attention for the use of acronyms in all abstract and in the whole of the manuscript. Methods: clarify who are the two groups of patients and the timeline that will be done the assessments or follow-ups. Which data collection instruments were used and how you got the subjects (inclusion and exclusion criteria).
Introduction
In the 42nd line, must to describe the inhaler devices and the therapeutics for COPD patients and it’s importance for controlling the symptoms and the exacerbations. The objective 2 (O2) in 96th line its not well define. Perhaps, To assess compliance…..and next the justification (like you state in O1).
Materials and Methods:
You need to define more clearly the hypotheses by presenting the significant relationship between which variables and only after state the literature that you support for those.
How the physicians were able to establish randomness among the patients who enter for an appointment? They will first have to check the inclusion and exclusion criteria and only then ask the questions to the patients. Explain further this eligibility and data collection process.
What is the difference between the inhaler recommendations for the recommendation group and the “type of device they would choose for each patient” in the CG (line 168)? If doctor advice an inhaler, it’s not the same for recommendation? Explain more clearly.
How you define errors in the technique of inhaler device? How did you observe these errors? You have an observation grid? It was only a doctor to observe? And about the subjectivity?
Results: the results are presented well by hypotheses defined.
Discussion: In the beginning of this topic, it is suitable present the main results of the study and after the discussion with other studies.
Conclusions: refer the importance of patient involvement in inhaler choice that was related to the frequency of errors.
Would the errors decrease only with the involvement of the patient? And about the family? And other health professionals? Wouldn't it be important to check the practices of physicians and other professionals (for example, nurses) in health education on how to use these devices? And establish a frequency of consultations to monitor patients' abilities? These are questions to consider in future investigations and reflections for improving health practices.
Author Response
- How the physicians were able to establish randomness among the patients who enter for an appointment? They will first have to check the inclusion and exclusion criteria and only then ask the questions to the patients. Explain further this eligibility and data collection process.
Answer: In order to reduce the multi-individual character of the doctor-patient relationship, we decided that the patients to be included in the study would be selected by a single investigating physician. Thus, patients who presented on their own initiative to the pulmonologist chosen as the investigating physician for the symptoms for which they presented for consultation were also checked for inclusion and exclusion criteria in the history. If these were met, the patient was presented with the option of being evaluated in or out of the study. Once agreement was obtained, the patient was placed in one of two groups - the recommended group or the chosen group. The first patient entered into the analysis was placed in the recommended group, the second patient who met the inclusion and exclusion criteria was placed in the chosen group. The process continued until data were successfully collected from 100 patients for each group.
- What is the difference between the inhaler recommendations for the recommendation group and the “type of device they would choose for each patient” in the CG (line 168)? If doctor advice an inhaler, it’s not the same for recommendation? Explain more clearly.
Answer: Subjects in the recommended group (RG) were presented with all three inhaler devices available in Romania for their stage of COPD - Genuair, Respimat and Breezhaler and from which the investigating physician would choose one. However, before the investigating physician decided on the type of device the patient would use during the study, patients were asked to indicate which type of device they would have chosen if they had been put in the position of deciding factor. On the patient follow-up chart, the physician noted which device type the patient in the RG would have opted for if given the opportunity. The treatment and inhaler device for patients in this group (RG) was recommended/chosen by the investigating pulmonologist.
Subjects in the choice group(CG) were provided with all three inhalation devices considered in the study - Genuair, Respimat and Breezhaler. While patients were reviewing the inhaler devices from which they were to choose one, the investigating pulmonologist noted on the patient's monitoring record which type of device they would have chosen. Patients were then asked to choose/decide for themselves which inhaler device they felt was most suitable for them, and which they would use during the study. It was the patients who chose the inhaler device in the CG.
- How you define errors in the technique of inhaler device? How did you observe these errors? You have an observation grid? It was only a doctor to observe? And about the subjectivity?
Answer: As mentioned in the Material and Method chapter, we decided that the interaction with the subjects entered into the study would be carried out by a single investigating physician. In this way we tried to minimize subjectivity on errors in patient inhalation technique and to maintain the same standard of training on correct inhalation technique for each of the three devices examined in the study.
We considered the inhalation technique used by the patient to be wrong if we identified at least one wrong element in the process of administering the inhalation treatment, starting from: preparation of the inhaler device (fitting the reservoir in the Respimat device or piercing the cap in the case of the Breezhaler device), loading the dose to be inhaled(putting the caps in the Breezhaler inhaler device, absence of pressing the flap in the case of the Genuair device or absence of cocking the Respimat device), absence of full exhalation before inhalation, absence of deep inspiration during inhalation, absence of apnea after inhalation and rapid and sudden exhalation after inhalation of the treatment. As mentioned in the limit chapter, even though we used the same investigating physician in the process of instructing patients on correct inhalation technique, as well as in the process of assessing the correctness of inhalation technique, we believe that subjectivity in the assessment of inhalation technique cannot be completely eliminated.
Thank you!